# Inhibitory Investigations of Acyl-CoA Derivatives against Human Lipoxygenase Isozymes

**DOI:** 10.3390/ijms241310941

**Published:** 2023-06-30

**Authors:** Michelle Tran, Kevin Yang, Alisa Glukhova, Michael Holinstat, Theodore Holman

**Affiliations:** 1Department of Chemistry and Biochemistry, University of California Santa Cruz, Santa Cruz, CA 95064, USA; mimytran@ucsc.edu (M.T.); kyang55@ucsc.edu (K.Y.); 2Department of Biochemistry and Pharmacology, University of Melbourne, Melbourne, VIC 3010, Australia; glukhova.a@wehi.edu.au; 3Walter and Eliza Hall Institute of Medical Research, Parkville, VIC 3052, Australia; 4Department of Medical Biology, University of Melbourne, Melbourne, VIC 3010, Australia; 5Drug Discovery Biology, Monash Institute of Pharmaceutical Sciences, Monash University, Parkville, VIC 3052, Australia; 6ARC Centre for Cryo-Electron Microscopy of Membrane Proteins, Monash Institute of Pharmaceutical Sciences, Monash University, Parkville, VIC 3052, Australia; 7Department of Pharmacology, University of Michigan Medical School, Ann Arbor, MI 48109, USA; mholinst@med.umich.edu

**Keywords:** lipoxygenase, acyl-coenzyme A, inhibition

## Abstract

Lipid metabolism is a complex process crucial for energy production resulting in high levels of acyl-coenzyme A (acyl-CoA) molecules in the cell. Acyl-CoAs have also been implicated in inflammation, which could be possibly linked to lipoxygenase (LOX) biochemistry by the observation that an acyl-CoA was bound to human platelet 12-lipoxygenase via cryo-EM. Given that LOX isozymes play a pivotal role in inflammation, a more thorough investigation of the inhibitory effects of acyl-CoAs on lipoxygenase isozymes was judged to be warranted. Subsequently, it was determined that C18 acyl-CoA derivatives were the most potent against h12-LOX, human reticulocyte 15-LOX-1 (h15-LOX-1), and human endothelial 15-LOX-2 (h15-LOX-2), while C16 acyl-CoAs were more potent against human 5-LOX. Specifically, oleoyl-CoA (18:1) was most potent against h12-LOX (IC_50_ = 32 μM) and h15-LOX-2 (IC_50_ = 0.62 μM), stearoyl-CoA against h15-LOX-1 (IC_50_ = 4.2 μM), and palmitoleoyl-CoA against h5-LOX (IC_50_ = 2.0 μM). The inhibition of h15-LOX-2 by oleoyl-CoA was further determined to be allosteric inhibition with a *K_i_* of 82 +/− 70 nM, an α of 3.2 +/− 1, a β of 0.30 +/− 0.07, and a β/α = 0.09. Interestingly, linoleoyl-CoA (18:2) was a weak inhibitor against h5-LOX, h12-LOX, and h15-LOX-1 but a rapid substrate for h15-LOX-1, with comparable kinetic rates to free linoleic acid (*k_cat_* = 7.5 +/− 0.4 s^−1^, *k_cat_/K_M_* = 0.62 +/− 0.1 µM^−1^s^−1^). Additionally, it was determined that methylated fatty acids were not substrates but rather weak inhibitors. These findings imply a greater role for acyl-CoAs in the regulation of LOX activity in the cell, either through inhibition of novel oxylipin species or as a novel source of oxylipin-CoAs.

## 1. Introduction

Fat and lipid metabolism is a complex process that involves the synthesis, breakdown, and transport of lipids throughout the body [1]. Lipids not only play a vital role in providing energy for the body but also help in the formation of cell membranes [2], hormone synthesis [3,4], and storage of fat-soluble vitamins [5]. Dysregulation of lipid metabolism has been associated with a range of diseases, including obesity [6,7], type 2 diabetes [8,9], and cardiovascular disease [10,11]. One important aspect of lipid metabolism is the synthesis and breakdown of fatty acids, the building blocks of triglycerides, phospholipids, and cholesterol esters [12,13]. Fatty acid metabolism involves a series of enzymatic reactions that convert fatty acids into acetyl-CoA, which can then enter the citric acid cycle for energy production [14,15,16]. In addition to their role in energy production, fatty acid-CoA esters have been shown to act as signaling molecules [17,18], regulating the activity of enzymes involved in lipid metabolism and other cellular processes [19,20]. Some studies have suggested that acyl-CoA synthase can even modulate the activity of cyclooxygenase (COX) and lipoxygenase (LOX) isozymes, which are involved in the production of pro-inflammatory lipid mediators [21]. For example, it has been observed that COX can be inhibited by palmitoyl-CoA, which can have concentrations as high as 10 µM in platelets [22]. Since COX reacts with fatty acids, such as arachidonic acid (AA), to produce specialized pro-resolving lipid mediators (SPMs) [23,24], this mechanism of inhibition could have significant biological effects. Lipoxygenase is also involved in the production of SPMs and uses the same substrate profile to produce potent pro- and anti-inflammatory mediators involved in allergic reactions and other immune responses [25,26]; therefore, inhibition of LOX isozyme by acyl-CoAs could have significant biological effects.

Lipoxygenases are a class of non-heme iron-containing dioxygenases that catalyze the oxygenation of polyunsaturated fatty acids, resulting in the formation of lipid mediators such as leukotrienes and lipoxins [27,28,29]. The high-resolution cryo-EM structures of human 12S-lipoxygenase (h12-LOX) were determined for the first time in a recent study, providing valuable insights into its oligomeric states, but equally interesting was the discovery that an acyl-CoA was bound to its active site [30]. The resolution of the h12-LOX structure was not high enough to identify the acyl-CoA; however, screening a family of acyl-CoAs showed that oleoyl-CoA was the most likely candidate due to its potency (IC_50_ = 32 µM). The current work expands on this result, screening a larger family of acyl-CoAs against h5-LOX, h12-LOX, h15-LOX-1, and h15-LOX-2. This work provides new insights into the inhibitory mechanism of acyl-CoAs against LOX isozymes, potentially indicating an endogenous role for acyl-CoAs as LOX inhibitors, which could lead to the development of novel therapeutics targeting these enzymes.

## 2. Results and Discussion

### 2.1. IC_50_ Determination of Unsaturated Fatty Acyl-CoAs

For h12-LOX, the IC_50_ values for palmitoyl-CoA (16:0), palmitoleoyl-CoA (16:1), stearoyl-CoA (18:0), linoleoyl-CoA (18:2), ɣ-linolenoyl-CoA (18:3), and docosahexaenoyl-CoA (22:6) indicated weak inhibition, with values greater than 200 µM (Table 1). Oleoyl-CoA (18:1) and arachidonoyl-CoA (20:4) had slightly greater inhibition potency, with IC_50_ values of 32 ± 4 µM and 110 ± 20 µM, respectively (Table 1). These data have been published previously [30].

For h15-LOX-1, the IC_50_ values for palmitoyl-CoA (16:0), palmitoleoyl-CoA (16:1), linoleoyl-CoA (18:2), ɣ-linolenoyl-CoA (18:3), arachidonoyl-CoA (20:4), and docosahexaenoyl-CoA (22:6) indicated weak inhibition, with all values being greater than 50 µM (Table 1). However, oleoyl-CoA (18:1) and stearoyl-CoA (18:0) had relatively stronger inhibition, with values of 39 ± 2 µM and 4.2 ± 0.6 µM, respectively. The data indicate that C18 acyl-CoAs have the greatest potency; however, since ɣ-linolenoyl-CoA (18:3) is not inhibitory, the degree of saturation also affects potency.

For h15-LOX-2, the IC_50_ values for palmitoyl-CoA (16:0), palmitoleoyl-CoA (16:1), linoleoyl-CoA (18:2), ɣ-linolenoyl-CoA (18:3), arachidonoyl-CoA (20:4), and docosahexaenoyl-CoA (22:6) indicated weak inhibition, with values greater than 100 μM (Table 1). Stearoyl-CoA (18:0) had relatively strong inhibition, with a value of 7.6 ± 1 µM; however, oleoyl-CoA (18:1) had exceptionally strong inhibition, with an IC_50_ of 620 ± 60 nM. Similar to what was seen for h15-LOX-1, the C18 acyl-CoAs had the greatest potency; however, with the presence of three double bonds, as with ɣ-linolenoyl-CoA (18:3), all inhibitor potency was lost.

For h5-LOX, the IC_50_ values for ɣ-linolenoyl-CoA (18:3) and docosahexaenoyl-CoA (22:6) indicated weak inhibition, with values greater than 200 µM. Stearoyl-CoA (18:0), oleoyl-CoA (18:1), and linoleoyl-CoA (18:2) had slightly stronger inhibition, with IC_50_ values greater than 50 µM (Table 1). However, palmitoyl-CoA (16:0) and palmitoleoyl-CoA (16:1) had low micromolar inhibition, with values of 3.3 ± 0.3 µM and 2.0 ± 0.4 µM. The C16 acyl-CoAs’ potency against h5-LOX contrasts with the other LOX isozymes and is possibly due to the active site requirement for the reverse binding of the substrate of h5-LOX relative to the other isozymes.

In summary, the majority of the acyl-CoAs were not inhibitors against LOX isozymes; however, certain acyl-CoAs displayed micromolar potency. For h5-LOX, the C16 acyl-CoAs displayed the most promising inhibitory properties, with low micromolar potencies. However, with all the other isozymes—h15-LOX-1, h15-LOX-2, and h12-LOX—the C18 acyl-CoAs had the most potent inhibitory properties. Interestingly, adding double bonds to the acyl chain, such as from oleoyl-coA (18:1) to linoleoyl-CoA (18:2), eliminated significant inhibition potency, and complete loss of inhibitory activity was observed with γ-linolenoyl-CoA (18:3). This effect was most likely due to the added restriction on structural freedom with the additional double bonds, which would decrease flexibility and lead to lower binding affinity, thus reducing the availability of the proper orientation. With respect to biological relevancy, the most potent acyl-CoA interaction was that between oleoyl-CoA and h15-LOX-2, which displayed submicromolar potency. This level of potency drew us to undertake a more detailed kinetic analysis (*vide infra*).

### 2.2. Acyl-CoA Substrate Activity

Considering that some of these acyl-CoAs could potentially be substrates for LOX isozymes, each acyl-CoA was tested at 70 µM against each LOX isozyme to establish the possibility of catalysis. These measurements revealed that none of the acyl-CoAs were substrates against the LOX isozymes, except for linoleoyl-CoA (18:2) and h15-LOX-1, as monitored by a change in absorbance at 234 nm (Table 2). The kinetic parameters were determined to be 7.5 ± 0.4 s^−1^ for *k_cat_*, 12 ± 0.8 µM for *K_M_*, and 0.62 ± 0.1 µM^−1^ s^−1^ for *k_cat_/K_M_* at 22 °C. These kinetic parameters are comparable to that of free linoleic acid, with *k_cat_* being 1.3-fold less and *k_cat_/K_M_* being 7-fold less than the published values [31], suggesting it as a possible biological substrate for h15-LOX-1.

### 2.3. IC_50_ Determination and Substrate Activity of Methyl Ester Unsaturated Fatty Acids

The acyl-CoAs are large ester derivatives of fatty acids and, given this fact, it was deemed appropriate to determine if simpler esters could also inhibit the LOX isozymes. Thus, methyl-13Z,16Z-docosadienoic, methyl 13(Z),16(Z),19(Z)-docosatrienoate, and methyl docosatetraenoate were investigated and determined to be weak inhibitors, with IC_50_ values greater than 200 µM for h12-LOX, h-15LOX-1, and h15-LOX-2. It should be noted that none of the methyl ester unsaturated fatty acids were substrates, except for methyl ester arachidonic acid and h15-LOX-1, as previously observed. However, the rate of this activity was low enough to still allow for IC_50_ measurements.

### 2.4. Kinetic Investigation of Oleoyl-CoA (18:1) Inhibition of h15-LOX-2

Oleoyl-CoA displayed nanomolar IC_50_ potency against h15-LOX-2 (620 ± 60 nM), which led to further kinetic investigations at varying inhibitor concentrations. The replot of the apparent *K_M_* versus concentration for oleoyl-CoA exhibited a hyperbolic response with increasing amounts of acyl-CoA from 624 nM to 3.6 μM (Figure 1). This hyperbolic response is representative of allosteric inhibition (i.e., partial inhibition), which was previously observed with 13-(S)-HOTrE(γ) and h15-LOX-2, as described in Figure 1 (Equations (1)–(4)) [32].
(1)$$\begin{array}{c} 1/v=(\alpha K_{M}/k_{\mathrm{cat}})[([I]+K_{i})/(\beta [I]+\alpha K_{i})]\times 1/[S]\\ +1/k_{\mathrm{cat}}\times [([I]+\alpha K_{i})/(\beta [I]+\alpha K_{i})] \end{array}$$
(2)$$K_{M}(\mathrm{app})=\alpha K_{M}[([I]+K_{i})/([I]+K_{i})]$$
(3)$$k_{\mathrm{cat}}/K_{M}=(k_{\mathrm{cat}}/\alpha K_{M})[(\beta [I]+\alpha K_{i})/([I]+K_{i})]$$
(4)$$k_{\mathrm{cat}}=k_{\mathrm{cat}}[(\beta [I]+\alpha K_{i})/([I]+\alpha K_{i})]$$

Utilizing the equations derived from Figure 1, the *K_M_* (app) values were fitted with Equation 2 to yield a value for α of 3.2 +/− 1 and a value for K_i_ of 82 +/− 70 nM (Figure 1). These two parameters were then utilized to fit Equation (3) relative to *k_cat_*/*K_M_* (app), which yielded a β value of 0.26 +/− 0.23 (Figure 2). Given the high error for the β value, β was also determined by applying the values for α and K_i_ from the *K_M_* (app) plot to Equation 4 and fitted to the *k_cat_* data (Figure 3). This fit yielded a β value of 0.30 +/− 0.07, consistent with the Equation 3 fit but with a lower error value. These kinetic values indicate hyperbolic allostery, α > 1 (K-type inhibition) and β < 1 (V-type inhibition), with the kinetic change being equally observed in both *K_M_* (α = 3.2 +/− 1) and *k_cat_* (β = 0.30 +/− 0.07). This combined effect is emphasized in the β/α value, the allosteric effect on *k_cat_/K_M_*, which is significantly less than 1 (β/α = 0.09) and indicates *k_cat_/K_M_* allosteric inhibition. These allosteric kinetic data indicate the formation of a catalytically active ternary complex (I·E·S) between h15-LOX-2 and oleoyl-CoA and are consistent with our previous finding of an allosteric site for 13-(S)-HODE with 15-LOX-2 [32]. It should be noted that the K_i_ values from our hyperbolic fits (82 +/− 70 nM) were dramatically lower than those determined from IC_50_ measurements (620 ± 60 nM), which is consistent with the more accurate nature of the hyperbolic fits relative to the IC_50_ fits, due to the oversimplification of the IC_50_ analysis.

Interestingly, in our recently published structure [30], oleoyl-CoA was bound in the active site of 12-LOX with its head group occupying the entrance to the binding site and the fatty acid tail extending deep into the U-shaped catalytic cavity. Similar to the previously predicted pose of AA bound to h12-LOX [33], H596 forms hydrogen bonds with the carbonyl group of the thioester bond. This is also consistent with the predicted pose of AA bound to h12-LOX, where L407, F414, A417, and V418 form Van der Waals interactions with the oleic acid tail. The h12-LOX binding site is quite narrow, consisting of two mostly linear segments connected by a single kink near the C9 position of the oleic acid (equivalent to C11 of AA). The shape of the binding cavity likely restricts the binding of the other acyl-CoAs tested in this study due to the strict requirements imposed on the position and saturation state of the fatty acid. With respect to the allosteric inhibition observed for h15-LOX-2, its binding site and that of h5-LOX and h15-LOX-1 appear to be wider and shorter, providing a possible explanation for their slightly better tolerance of different fatty acid saturation states.

## 3. Materials and Methods

### 3.1. Chemicals

Fatty acids used in this study were purchased from Nu Chek Prep, Inc. (Elysian, MN, USA). All other solvents and chemicals were reagent-grade or better and were used as purchased without further purification.

### 3.2. Synthesis of DHA-CoA

Reactions took place in an oxygen-deficient environment. DHA was added to an equimolar amount of butylated hydroxytoluene. Oxalyl chloride, along with 1 drop of dry dimethylformamide (DMF), was then added, and the mixture was stirred for 1 h at 37 °C and dried under vacuum. CoA was then added to a solution of tetrahydrofuran and 150 mM sodium bicarbonate in a ratio of 2.2/1.0. This solution was combined with the DHA solution for 30 min at 37 °C. The solution was dried to remove the tetrahydrofuran. Dichloromethane with 1.3% perchloric acid was used to extract the desired product away from the unreacted DHA acid. The water layer was retained and mixed with an equal volume of acetonitrile and lyophilized. The solid was washed with dry acetone and ether (Figure 2). The synthesis was based on published protocols [34,35].

### 3.3. Expression and Purification of h5-LOX, h12-LOX, h15-LOX-1, and h15-LOX-2

Overexpression and purification of his-tagged wild-type h12-LOX (Uniprot entry P18054) [36], h15-LOX-1 (Uniprot entry P16050) [36], and h15-LOX-2 (Uniprot entry O15296) [37] were performed using nickel-affinity chromatography. The purity of h12-LOX, h15-LOX-1, and h15-LOX-2 was assessed using SDS gel as greater than 85%, and metal content was assessed with a Finnigan inductively coupled plasma mass spectrometer (ICP-MS, Thermo Fischer Scientific, San Jose, USA) via comparison with iron standard solution. Cobalt-EDTA was used as an internal standard. WT h5-LOX (UniProt entry P09917) was expressed in Rosetta 2 cells (Novagen) transformed with the pET14b-Stable-5-LOX plasmid (a gift from Marcia Newcomer of Louisiana State University) and grown in Terrific Broth containing 34 μg/mL chloramphenicol and 100 μg/mL ampicillin at 37 °C for 3.5 h, then placed at 20 °C for an additional 26 h. Cells were pelleted and resuspended in 50 mM Tris (pH 8.0), 500 mM NaCl, 20 mM imidazole with 1 μM pepstatin, 100 μM PMSF, and DNaseI (Sigma, Kanagawa, Japan). The cells were lysed in a French pressure cell and centrifuged at 40,000× *g* for 20 min at 4 °C. Saturated ammonium sulfate was added to achieve a final solution of 10% and inverted several times. The lysate was centrifuged at 40,000× *g* for 10 min at 4 °C. The pellet was discarded, and the supernatant was increased to a final concentration of 50% ammonium sulfate. The supernatant was rotated for 10 min at 4 °C then centrifuged at 40,000× *g* for 10 min at 4 °C. The pellet was split into 150 mg aliquots. The WT h5-LOX used for the kinetics in this work was not purified due to a dramatic loss in activity and was therefore prepared as an ammonium-sulfate precipitate.

### 3.4. IC_50_ Determination

IC_50_ values for each acyl-CoA were measured against h5-LOX, h12-LOX, h15-LOX-1, and h15-LOX-2. The reactions were carried out in 25 mM HEPES buffer (pH 8.00), 0.01% Triton X-100, and 10 μM AA for h12-LOX; 25 mM HEPES buffer (pH 7.5), 0.01% Triton X-100, and 10 μM AA for h15-LOX-1 and h15-LOX-2; and 50 mM HEPES buffer (pH 7.5), 100 µM EDTA, 50 mM NaCl, 200 µM ATP, and 10 μM AA for h5-LOX. IC_50_ reactions were implemented with h15-LOX-1 (0.125 μM), h12-LOX (0.300 μM), h15-LOX-2 (0.5 μM), and h5-LOX (∼600 nM ammonium sulfate salt) at 22 °C in a 1 cm^2^ quartz cuvette containing 2 mL of the according buffer and 10 μM AA for no longer than 2 min to avoid any cleavage of the acyl-coenzyme A. Inhibitor concentrations ranging from 50 μM to 0.01 μM were used. All reactions were conducted using a Cary-UV Vis spectrophotometer (Agilent, Santa Clara, CA, USA). The fastest rates over 15 s intervals were recorded. IC_50_ values were obtained by determining the percent of inhibition of the enzymatic rate at ten inhibitor concentrations and plotting the rate against the inhibitor concentration, followed by a hyperbolic saturation curve fit. The saturation curve fits were performed in duplicate or triplicate, depending on the quality of the data. It should be noted that stability measurements of the acyl-CoAs indicated that there was no appreciable hydrolysis in the short timeframe of the kinetic measurements (approximately one minute). Through HPLC monitoring, we were able to determine a lack of degradation of the fatty acyl-CoAs over the first hour.

### 3.5. Enzymatic Product Determination

h15-LOX-2 (0.200 μM), h15-LOX-1 (0.125 μM), h5-LOX (0.300 μM), and h12-LOX (0.300 μM) were reacted with 30 μM of acyl-CoAs in 2 mL of 25 mM HEPES (pH 7.5 for h15-LOX-1 and h15-LOX-2, pH 8.0 for h12-LOX) at room temperature with ambient oxygen. The same buffer as used to determine IC_50_ values for h5-LOX was used for product determination. Turnover was monitored by absorbance at 234 nm.

### 3.6. Kinetic Investigation of Oleoyl-CoA (18:1) Inhibition of h15-LOX-2

The reactions were carried out in 25 mM HEPES buffer (pH 7.5), 0.01% Triton X-100. Varying amounts of substrates from AA ranging from 50 μM to 0.01 μM and 0, 150, 300, 450, and 600 nM of 18:2 acyl-coA were used. Reactions were performed at 22 °C in a 1 cm^2^ quartz cuvette containing 2 mL of the according buffer for no longer than 2 min to avoid any cleavage of the acyl-coenzyme A. Using a Cary-UV Vis spectrophotometer, the fastest rates over 15 s intervals were recorded. Kinetic values were obtained by plotting the rate against the substrate concentration, followed by a hyperbolic saturation curve fit. The saturation curve fits were performed in triplicate and on separate days to ensure there was no bias. The equations used to find kinetic values were from previous studies [32].

## 4. Conclusions

In conclusion, our previously reported cryo-EM structure for h12-LOX indicated an acyl-CoA bound to its active site, most likely oleoyl-CoA [30]. Based on this discovery, we investigated the inhibitory potency of acyl-CoAs against lipoxygenase isozymes and were able to determine that acyl-CoAs can be micromolar inhibitors against LOX isozymes, with oleoyl-CoA (18:1) being the most potent, having nanomolar allosteric potency against h15-LOX-2. Surprisingly, LA-CoA is a rapid substrate for h15-LOX-1, suggesting its product could have biological activity. Considering that acyl-CoAs exist in the cell at micromolar concentrations [38], it is reasonable to assume that acyl-CoAs could have a significant effect on the cellular activity of LOX isozymes, both as inhibitors and substrates. We are currently investigating whether the cellular concentrations of acyl-CoAs affect LOX activity in vivo and if this activity affects LOX signaling.

## Data Availability

No archived data for this work. All data presented.

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
