# Peer review of "Inhibitory Investigations of Acyl-CoA Derivatives against Human Lipoxygenase Isozymes"

_ijms, 2023, doi:10.3390/ijms241310941_

Round 1

Reviewer 1 Report

The manuscript entitled "Inhibitory Investigations of Acyl-CoA derivatives against Human Lipoxygenase Iso-1 zymes " provides an overview about inhibitory role of acyl-CoA derivatives against LOX, Lipoxygenase. Authors purchased the Acyl-CoA derivatives and measured the IC50 against h5-LOX, h12-LOX, h15-LOX-1, 133 and h15-LOX-2 and showed that certain acyl-CoAs are inhibitors against LOX isozymes in micromolar concentration, however Oleoyl-CoA(18:1) being the most potent inhibitor against h15-LOX-2 with IC50 values in nanomolar concentration.

Overall, the abstract is clear and can stand alone. The introduction summarizes the topic well.  The methods are appropriate, detailed and the analysis looks reasonable for the inhibitory activity. The results are presented clearly and accurately and match the methods. Below are some of my major and minor concerns regarding the article and experiments:

Major Concern:

1.     Page 4, Authors performed the inhibition kinetics in duplicate or triplicate to fit the curve and calculate the IC50, it would be great if the authors can provide fitting curves so that one can look into the STD values and fitting of the plot. This will ensure better transparency and understanding of the inhibition.

2.     Authors used Acyl-CoA derivatives to measure the inhibitory effect against hLOX, it seems that the binding affinities of Acyl-CoA derivatives varies with the LOX isozymes which really begs the question, did authors tried to look at the active site differences at the primary and 3D structural level? If yes what are the critical differences made the different IC50 values against LOX isozymes. As it was mentioned that acyl-CoA bound h12LOX structure available (Cryo-EM).

Minor concern:

1.     Page 11, Line 220: consider rephrasing - ‘For h5-LOX, the C16 acyl-CoAs displayed low micromolar potency, but for h15-LOX-1’.

2.     Page 9, Overall, the conclusion would benefit from small revisions for conciseness and importance of the AcylCoA as an inhibitor?

3.     Authors used the Kcat and substrate concentration for plotting the hyperbolic saturation curve, It would be great if the authors can provide the concentration of LOX isozyme used for performing enzyme kinetics?

Author Response

The manuscript entitled "Inhibitory Investigations of Acyl-CoA derivatives against Human Lipoxygenase Iso-1 zymes " provides an overview about inhibitory role of acyl-CoA derivatives against LOX, Lipoxygenase. Authors purchased the Acyl-CoA derivatives and measured the IC50 against h5-LOX, h12-LOX, h15-LOX-1, 133 and h15-LOX-2 and showed that certain acyl-CoAs are inhibitors against LOX isozymes in micromolar concentration, however Oleoyl-CoA(18:1) being the most potent inhibitor against h15-LOX-2 with IC50 values in nanomolar concentration.

Overall, the abstract is clear and can stand alone. The introduction summarizes the topic well.  The methods are appropriate, detailed and the analysis looks reasonable for the inhibitory activity. The results are presented clearly and accurately and match the methods. Below are some of my major and minor concerns regarding the article and experiments:

            Thank you for the supportive comments.

Major Concern:

  1. Page 4, Authors performed the inhibition kinetics in duplicate or triplicate to fit the curve and calculate the IC50, it would be great if the authors can provide fitting curves so that one can look into the STD values and fitting of the plot. This will ensure better transparency and understanding of the inhibition.

We have added a file of supporting information with the graphs of the potent inhibitors. For the weak inhibitors, only one inhibitor concentration in triplicate was utilized, so no fits were obtained.

  1. Authors used Acyl-CoA derivatives to measure the inhibitory effect against hLOX, it seems that the binding affinities of Acyl-CoA derivatives varies with the LOX isozymes which really begs the question, did authors tried to look at the active site differences at the primary and 3D structural level? If yes what are the critical differences made the different IC50 values against LOX isozymes. As it was mentioned that acyl-CoA bound h12LOX structure available (Cryo-EM).

We did not computer dock the FA-CoAs into the active sites of all four LOX isozymes. However, we do have the cryo-EM structure of 12LOX with a FA-CoA bound (publication submitted to Blood). Therefore, we have added information on page 14 discussing this binding interaction and postulate possible reasons for the SAR data.

Minor concern:

  1. Page 11, Line 220: consider rephrasing - ‘For h5-LOX, the C16 acyl-CoAs displayed low micromolar potency, but for h15-LOX-1’.

            We have changed the line to the following, “ For h5-LOX, the C16 acyl-CoAs displayed the most promising inhibitory properties having low micromolar potencies. However, with all the other isozymes, h15-LOX-1. h15-LOX-l, and h12-LOX, the C18 acyl-CoAs had the most potent inhibitory properties.”

  1. Page 9, Overall, the conclusion would benefit from small revisions for conciseness and importance of the AcylCoA as an inhibitor?

            We have rewritten the conclusions so that they are more concise.

  1. Authors used the Kcatand substrate concentration for plotting the hyperbolic saturation curve, It would be great if the authors can provide the concentration of LOX isozyme used for performing enzyme kinetics?

            This info was provided in the materials section (Page 7).

Reviewer 2 Report

The submission by Michelle Tran and co-workers entitled " Inhibitory Investigations of Acyl-CoA Derivatives against Human Lipoxygenase Isozymes" implicate a greater role of acyl-CoAs in the regulation of LOX activity. The article represents a well-structured study, which is an important contribution to biochemical and pharmacological research. However, the quality of printed version of the manuscript is not satisfactory. Some tables and figures are not placed appropriately and overlay the text. I guess, it might be simply a technical problem. I had the feeling that the manuscript needed an additional proof reading by the authors since it contained numerous typos as listed below:

Line 54: Two points at the end of the first sentence;

Line 60: The point at the end of the sentence is missing;

Line 106: I guess, the sentence “The solution was rotovated..” contais a misprint;

Line 123: The sentence “…DNaseI (2 Kunitz/g) (Sigma)…” contains a misprint;

Line126: The sentence “ The lysis was centrifuged” seems to be not correct. It should be changed to “The lysate was centrifuged”;

Lines 133, 142, 168, 169, 175 etc: Please use a subscript option for numbers when typing IC50;

Lines 187-188: The sentence „.. but the added unsaturation of É£-linolenoyl-CoA(18:3) lost all potency“ is too awkward and need to be rephrased;

Line 195: please, make the plural form for “the other LOX isoenzymes”;

Lines 220-221: The sentence „.. For h5-LOX, the C16 acyl-CoAs displayed low micromolar potency, but for h15-LOX-1.” seems not to be complete;

Line 222: the term „.. adding unsaturation   “ sounds too awkward;

Line 234: the point is missing at the end of the sentence;

Line 302: the reference, which addresses „previous finding of an allosteric site for 13-(S)-HODE with 15-LOX-2“ is missing.

Line 317: please use low case for Oleoyl-CoA

Major:

The table 3 is not informative and should be omitted. IC50 values presented for different compounds and different isoenzymes are identical!

 The authors make an interesting finding that introduction of one additional double bond as shown for É£-linolenoyl-CoA(18:3) lead to the lost C18 acyl-CoAs inhibitory potency. What was the reason for this? The authors cjuld at least discuss this issue in the revised version of the manuscript.

Author Response

Reviewer 2

The submission by Michelle Tran and co-workers entitled " Inhibitory Investigations of Acyl-CoA Derivatives against Human Lipoxygenase Isozymes" implicate a greater role of acyl-CoAs in the regulation of LOX activity. The article represents a well-structured study, which is an important contribution to biochemical and pharmacological research. However, the quality of printed version of the manuscript is not satisfactory. Some tables and figures are not placed appropriately and overlay the text. I guess, it might be simply a technical problem. I had the feeling that the manuscript needed an additional proof reading by the authors since it contained numerous typos as listed below:

            We apologize for these problems. We will work with the editor to fix these problems.

Line 54: Two points at the end of the first sentence;

            This has been corrected.

Line 60: The point at the end of the sentence is missing;

            This has been corrected.

Line 106: I guess, the sentence “The solution was rotovated..” contais a misprint;

            This has been corrected.

Line 123: The sentence “…DNaseI (2 Kunitz/g) (Sigma)…” contains a misprint;

            This has been corrected.

Line126: The sentence “ The lysis was centrifuged” seems to be not correct. It should be changed to “The lysate was centrifuged”;

            This has been corrected.

Lines 133, 142, 168, 169, 175 etc: Please use a subscript option for numbers when typing IC50;

            This has been corrected.

Lines 187-188: The sentence „.. but the added unsaturation of É£-linolenoyl-CoA(18:3) lost all potency“ is too awkward and need to be rephrased;

We have rewritten this sentence to “Similar to that seen for h15-LOX-1, the C18 acyl-CoAs had the greatest potency. However, with the presence of three double bonds, as with É£-linolenoyl-CoA (18:3),  all inhibitor potency was lost.”

Line 195: please, make the plural form for “the other LOX isoenzymes”;

            This has been corrected.

Lines 220-221: The sentence „.. For h5-LOX, the C16 acyl-CoAs displayed low micromolar potency, but for h15-LOX-1.” seems not to be complete; 

We have rewritten this section.

Line 222: the term „.. adding unsaturation   “ sounds too awkward;

We have rewritten this section.

Line 234: the point is missing at the end of the sentence;

            This has been corrected.

Line 302: the reference, which addresses „previous finding of an allosteric site for 13-(S)-HODE with 15-LOX-2“ is missing.

The reference has been added.

Line 317: please use low case for Oleoyl-CoA

            This has been corrected.

Major:

The table 3 is not informative and should be omitted. IC50 values presented for different compounds and different isoenzymes are identical!

            We have omitted Table 3.

 The authors make an interesting finding that introduction of one additional double bond as shown for É£-linolenoyl-CoA (18:3) lead to the lost C18 acyl-CoAs inhibitory potency. What was the reason for this? The authors could at least discuss this issue in the revised version of the manuscript.

This is an excellent suggestion. We have rewritten this section and added the suggested discussion points.

Reviewer 3 Report

The present manuscript was fine. There were some minor comments.

In Abstract, Mobbs et al. was cited. Was not the number of reference needed? Or, was the citation needed in Abstract? The reference [30] in the present manuscript was Mobbs et al.

What is the abbreviation "DMF" in Methods?

In the review version, figure numbers and figure legends were lost. It was not clear which were figures 1-3.   

Author Response

Reviewer 3

The present manuscript was fine. There were some minor comments.

In Abstract, Mobbs et al. was cited. Was not the number of reference needed? Or, was the citation needed in Abstract? The reference [30] in the present manuscript was Mobbs et al.

            We have corrected this and simply added the reference number.

What is the abbreviation "DMF" in Methods?

            We have now defined DMF.

In the review version, figure numbers and figure legends were lost. It was not clear which were figures 1-3.   

            We apologize for this omission, we will work with the editor to fix this problem.